# Preparation and Characterization of Soft-Hard Block Copolymer of 3,4-IP-*b*-*s*-1,2-PBD Using a Robust Iron-Based Catalyst System

**DOI:** 10.3390/polym16081172

**Published:** 2024-04-21

**Authors:** Yingnan Zhao, Shiliang Xu, Yao Yu, Heng Liu, Feng Wang, Lihua Na, Qi Yang, Chunyu Zhang, Xuequan Zhang

**Affiliations:** 1Key Laboratory of Advanced Rubber Material, Ministry of Education, Qingdao University of Science and Technology, Qingdao 266042, China; ironcatalyst@163.com (Y.Z.); x17860771919@163.com (S.X.); b021020003@mails.qust.edu.cn (Y.Y.); hengliu@qust.edu.cn (H.L.); lhna@qust.edu.cn (L.N.); 03715@qust.edu.cn (Q.Y.); xqzhang@qust.edu.cn (X.Z.); 2Shandong Provincial College Laboratory of Rubber Material and Engineering, School of Polymer Science and Engineering, Qingdao University of Science & Technology, Qingdao 266042, China; 3Key Lab of Rubber-Plastics, Ministry of Education/Shandong Provincial Key Lab of Rubber-Plastics, School of Polymer Science and Engineering, Qingdao University of Science & Technology, Qingdao 266042, China

**Keywords:** iron-based catalyst, 3,4-polyisoprene, soft–hard block copolymer, *syndiotactic*-1,2-polybutadiene

## Abstract

A series of well-defined diblock copolymers, namely, 3,4-polyisoprene-*block*-*syndiotactic*-1,2-polybutadiene (3,4-PI-*b*-*s*-1,2-PBD), with a soft–hard block sequence were synthesized via an in situ sequential polymerization process using a robust iron-based catalytic system Fe(acac)_3_/(isocyanoimino) triptenylphosphorane (IITP)/Al*^i^*Bu_3_. This catalyst exhibits vigorous activity and temperature tolerance, achieving a polymerization activity of 5.41 × 10^6^ g mol_(Fe)_^−1^ h^−1^ at 70 °C with a [IP]/[Fe] ratio of 15,000. Moreover, the quasi-living polymerization characteristics of the catalyst were verified through kinetic experiments. The first-stage polymerization of isoprene (IP) is performed at 30 °C to give a soft 3,4-PI block, and then a quantitative amount of 1,3-butadiene was added in situ to the quasi-living polymerization system to produce a second hard *s*-1,2-PBD. The *s*-1,2-PBD segments in block copolymers display a rodlike morphology contrasting with the spherulitic morphology characteristic of *s*-1,2-PBD homopolymers. The precise tunability of the length of the soft and hard chain segments of these novel elastic materials with the feed ratio of IP and BD, endowing them with outstanding mechanical properties and excellent dynamic mechanical properties, which are expected to be promising high-performance rubber materials.

## 1. Introduction

Block copolymers, characterized by their distinct chain segment structures, encapsulate the properties of their constituent homopolymers and exhibit a superior performance compared to the corresponding homopolymer blends [1,2]. Significant advancements have been achieved in the fields of synthesis and characterization technologies in recent years [3,4]. Isoprene (IP) and 1,3-butadiene (BD) are fundamental diene monomers for industrial production and can be polymerized to access polymers with a diverse regio- and/or stereo-selectivity, such as *cis*-1,4, *trans*-1,4, *syndio*-, *atactic*-3,4, and/or 1,2-polyolefins. The microstructure of the resultant polymer significantly affects the properties and final application of materials. For instance, polybutadiene (PBD) and polyisoprene (PI) with >96% *cis*-1,4-configuration possess soft rubber properties [5,6,7,8], while gutta-percha with a high *trans*-1,4 selectivity has unique applications in insulating and medicinal materials [9,10,11]. It has been reported that amorphous 3,4-polyisoprene (3,4-PI) exhibited outstanding wet-skid and low rolling resistance, potential properties of materials for producing future high-performance tires [12,13,14,15]. Meanwhile, *syndiotactic* 1,2-polybutadiene (*s*-1,2-PBD), a high-strength thermoplastic elastomer with excellent wear resistance and aging resistance, has been widely used in films, pipes, hoses, and rubbers [16,17,18]. Therefore, block copolymers incorporating soft (amorphous 3,4-PI) or hard (crystalline *s*-1,2-PBD) segments can effectively exploit the respective polymer’s benefits. This strategy aims to obtain innovative high-performance materials and broaden their potential applications.

In recent decades, co-ordination polymerization technology has enabled the construction of most block copolymers, as a result of advancements in catalysts and polymerization techniques. The sequential polymerization technique of introducing another monomer into the living polymer chain and the sequential polymerization technique of adding structural regulators to the living polymerization system provide two promising strategies for achieving soft–hard block copolymers [19,20]. For example, the co-ordination copolymerization of polar diphenylphosphine styrene (*p*-StPPh_2_) and IP was conducted using a half-sandwich scandium (C_5_Me_4_SiMe_3_) Sc (CH_2_C_6_H_4_NMe_2_-*o*)_2_ catalyst. Notably, p-StPPh_2_ was incorporated into the polymer chains after the almost complete depletion of IP, resulting in the synthesis of phosphine-functionalized 1,4/3,4-isoprene-styrene block copolymers [21]. The recent study also presents examples of 3,4-PI-*b*-polycaprolactone block copolymers prepared through the sequential addition of hard *ε*-caprolactone to the living lutetium (Lu)−PI active species, followed by ring-opening polymerization [22]. Exploiting the inherent characteristic of cobalt-based catalysts and incorporating Lewis bases such as triphenylphosphine (PPh_3_) or diphenylcyclohexylphosphine (PCyPh_2_) as selective modulators, the *cis*-1,4-*b*-*s*-1,2 soft–hard block polybutadiene has been successfully synthesized [23,24].

Benefiting from their high metal abundance, low cost, cost-effectiveness, biocompatibility, ease of preparation, and stability under a wide range of polymerization conditions, iron-based catalysts are preferred for use. Previously, we reported the utilization of a Fe(2-EHA)_3_/Al*^i^*Bu_3_/DEP catalytic system to synthesize stereoblock polybutadiene (e-PBD-*b*-*s*-1,2-PBD), which consists of atactic and syndiotactic sequences with alternating hard and soft segments by adjusting the [Al]/[Fe] molar ratio [25]. However, the catalytic activity of this catalyst system towards IP monomers still needs to be improved. Limited examples of 3,4-PI and 1,2-PBD copolymers have been reported so far, just random copolymers of both [26,27,28,29]. This is primarily subject to two critical constraints as follows. Firstly, unlike vinyl monomer polymerization, the regulation of stereoselectivity in diene polymerization becomes more complicated and challenging in controlling the regio- and stereo-selectivity of active species. Secondly, few catalytic systems can promote BD/IP polymerization in a 1,2/3,4 selective manner, wherein co-ordinated monomers undergo migration to C_3_ locations of the allylic carbons of Fe-*η*^3^-π-allyl, resulting in the formation of 1,2/3,4 structural units instead of the general C_1_ position in 1,4-insertions [7,30,31]. Currently, anionic or some Ziegler–Natta catalysts can initiate the 1,2 selective polymerizations of BD [32,33,34,35], whereas only specific rare-earth and transition metal complexes activated with methyl aluminoxane (MAO), alkyl aluminum, and/or borate provide 3,4 selectivity for polymerization of IP [36,37,38,39,40,41]. Our previous research presents a robust catalyst, iron(III) acetylacetonate/alkyl aluminum/(isocyanoimino)/triphenylphosphorane (Fe(acac)_3_/AlR_3_/IITP), which exhibits quasi-living characteristics for BD polymerization and stably generate *s*-1,2-PBD with crystallinity ranging from 56.2% to 73.5% [42].

In this work, the polymerization behavior of IP using Fe(acac)_3_/Al*^i^*Bu_3_/IITP was investigated systematically. The catalyst proved to be advantageous for achieving a high selectivity towards 3,4-insertion (3,4 (+1,2) > 53.8%) during IP polymerization while exhibiting ultra-high activity and thermal tolerance. The resultant living iron-PI active species could further initiate the successive polymerization of BD to *s*-1,2 and selectively provide 3,4-PI-*b*-*s*-1,2-PBD block copolymers. The chain structure, thermal behaviors, aggregation morphology, and dynamic and static mechanical properties of the copolymers were carefully characterized and investigated by GPC, NMR, DSC, XRD, tensile testing, and DMA.

## 2. Materials and Methods

### 2.1. Materials

Iron(III) acetylacetonate (Fe(acac)_3_, 0.02 mol L^−1^), (isocyanoimino) triphenylphosphorane (IITP, 0.0125 mol L^−1^), diethyl phosphite (DEP, 0.05 mol L^−1^), and 2,2′-azobis (2-methylpropionitrile) (AIBN, 0.02 mol L^−1^) were purchased from Sinopharm Chemical Reagent Co., Ltd. (Shanghai, China) and diluted with toluene. Triisobutyl aluminum (Al*^i^*Bu_3_, TIBA), triethyl aluminum (AlEt_3_, TEA), and diisobutyl aluminum hydride (Al*^i^*Bu_2_H, TDBAH) were purchased from the Macklin and diluted to 1.0 mol L^−1^ solutions in toluene. Polymerization grade isoprene (IP) and 1,3-butadiene (BD) were supplied by Qingdao Ludong Gas Co., Ltd. (Qingdao, China). Among them, BD was refined by successively passing through two columns packed with activated molecular sieves (4 Å) and KOH, respectively, and IP was dried over CaH_2_, and then distilled before use. Hexane and toluene were distilled over sodium-benzophenone prior to use. Other reagents and solvents were commercially available and used without purification.

### 2.2. Polymerization Procedure

All the manipulations were performed under a dry nitrogen atmosphere. A detailed polymerization procedure (corresponding to Run 2, Table 1) is described as a typical example. A hexane solution of IP (2.3 mol L^−1^, 25.6 mL) was placed in an oxygen- and moisture-free ampoule capped with a rubber septum. Then, followed by Fe(acac)_3_, IITP and Al*^i^*Bu_3_ at designed ratios were charged sequentially into the above ampoule. The polymerization was performed at 50 °C for 4 h and terminated by adding 2.0 mL of acidified ethanol involving an antioxidant reagent 2,6-di-tert-butyl-4-methylphenol (BHT, 1.0 wt%). A large amount of ethanol was added and washed repeatedly to convert the active end group into an inactive salt that can be dissolved in ethanol or water. Finally, dry the obtained product to a constant weight under vacuum at 50 °C. The polymer yield was determined by gravimetry analysis. For kinetic studies, multiple dry nitrogen-filled ampoules were prepared and IP homopolymerized as described above. It was ensured that each ampoule contained the same amount of all components. Maintain the solution at the polymerization temperature and quench one of the polymerization reactions with acidified ethanol at a given time. For the preparation of block copolymers 3,4-PI-*b*-*s*-1,2-PBD (taking Run 20, Table 2 as an example), firstly, the living PI chain was prepared as described above, the concentration and volume (including hexane) of IP are 0.5 mol L^−1^ and 11.8 mol L^−1^, respectively. After polymerization at 30 °C for 3 h, another hexane solution of BD (0.5 mol L^−1^, 23.5 mL) was rapidly added. The polymerization proceeded for an additional 4 h. The precipitation and dry procedures of the block copolymer are the same as that of preparing PI homopolymer.

### 2.3. Characterization

The molecular weight (*M*_n_) and molecular weight distributions (PDI) of the polymers were determined at 130 °C by AGILENT 1260 infinity II high-temperature gel permeation chromatography (HT-GPC). HPLC grade 1,2,4-trichlorobenzene (TCB) containing BHT (0.05 wt%) was used as the mobile phase with a flow rate of 1.0 mL min^−1^. The calibration was performed by using polystyrene. The microstructure of the resultant polymers was determined by ^1^H NMR, ^13^C NMR, and ^1^H-^13^C HSQC monitored on a BRUKER AVANCE NEO 400 MHz NMR spectrometer at 120 °C in *o*-dichlorobenzene-d_4_ (*o*-C_6_D_4_Cl_2_) or 1,1,2,2-tetrachloroethane-d_2_ (C_2_D_2_Cl_4_) with TMS as internal standard. The thermal properties of polymers were measured through a NETZSCH DSC 3500 Sirius under the nitrogen atmosphere with a heating/cooling rate of 10 °C min^−1^. The X-ray diffraction (XRD) patterns of copolymer samples in the range 2*θ* = 5°~40° were obtained with a Rigaku D/MAX/2500PC diffractometer. Transmission electron microscopy (TEM) measurements were performed on a JEM-2100 (JEOL, Tokyo, Japan) with an accelerating voltage of 200 kV. To prepare polymer film for TEM, 0.5 wt% toluene solution of the sample was cast on the hot glycerol to evaporate the solvent and annealed at 120 °C for 5 min, and then the films were transferred to the surface of deionized water and collected with copper grids [43,44]. By ISO 37:2011 [45], tensile experiments of polymers were performed on a tensile tester (INSTRON-5869) using the dumbbell samples (20 mm × 4 mm × 1 mm) with a tensile rate of 50 mm min^−1^. The dynamic mechanical analysis (DMA) experiments were carried out using a TA Q800 DMA with a tensile mode using the rectangular specimens (15 mm × 4 mm × 1 mm). For DMA tests, the samples were heated from −60 to 80 °C at 3 °C min^−1^ with a frequency of 10 Hz.

## 3. Results and Discussion

### 3.1. Homopolymerization Behavior of IP

For diene polymerization systems mediated by Ziegler–Natta-type iron-based catalysts, the cocatalyst not only serves as an activator for generating active species, but also plays a vital role in the activity and microstructure composition of the resulting polymer due to its alkylation ability and special hindrance [33,46,47,48]. This article selects IITP, which combines the isocyano and phosphoryl groups, as the donor with the activity and 3,4 selectivity advantages, as shown in Appendix A. Firstly, the effects of alkyl aluminum type and feeding amount on IP polymerization behavior were investigated using Al*^i^*Bu_3_, AlEt_3_, and Al*^i^*Bu_2_H as cocatalysts (Run 1–9, Table 1). The results demonstrated that iron-based ternary catalysts using different alkyl aluminum cocatalysts exhibited a favorable catalytic activity for IP polymerization at [Al]/[Fe] = 20; all of them were able to reach yields of more than 95% within 4 h, yielding PI with a content of 3,4 (+1,2) > 50%. However, at [Al]/[Fe] = 10, the catalytic activities of the alkyl-aluminum-catalyzed iron-based ternary catalysts followed the order of Al*^i^*Bu_3_ > Al*^i^*Bu_2_H > AlEt_3_. This could be attributed to the strong reducing ability of Al*^i^*Bu_2_H and AlEt_3_, which tends toward the excessive reduction of the catalyst’s active site Fe_3+_ to an inactive low-valent one, resulting in a much lower polymer yield [33,47]. The variation in molecular weight observed between the Al*^i^*Bu_3_ and Al*^i^*Bu_2_H systems may be ascribed to the stronger chain transfer tendency of Al*^i^*Bu_2_H compared to Al*^i^*Bu_3_ [48]. This implies that the IITP can stabilize the active site of the catalyst with a remarkable stability in terms of selectivity towards the 3,4 isomer despite changes in the cocatalysts [42].

The effect of [IP]/[Fe] ratios on the polymerization of IP was further examined at a fixed [Al]/[Fe] ratio of 20 (Runs 2 and 10–12, Table 1). Elevating the [IP]/[Fe] ratios from 10,000 to 30,000, the yield was kept at a high value of more than 72.0%, without an apparent change in the 3,4 selectivity. Further increasing the [IP]/[Fe] ratio to 50,000, strikingly, the polymer yield can still be maintained above 40%. Such high activity is rare in the previously reported late transition metal catalytic systems, which suffer entirely from the activity loss at the high IP/Fe molar ratio. Moreover, with an increasing amount of IP, the monotonous increase of the molecular weight was observed, together with the relatively narrow PDI of the resultant polymers. It can be attributed to the fact that the chain transfer reaction can be ignored, and the catalytic system has the characteristics of quasi-living polymerization [35]. Furthermore, the slightly increased 1,2 selectivity also went along with the increasing monomer feeding; meanwhile, the *T*_g_ also had an increased trend.

The thermal stability of the formed catalyst was also investigated by varying the polymerization temperature from 0 to 70 °C. As shown in Appendix A, the iron-based catalytic system involved in IITP significantly increases its activity with elevated polymerization temperatures. Surprisingly, the polymerization achieved a conversion of 79.5% and an excellent activity of up to 5.41 × 10^6^ g mol_(Fe)_^−1^ h^−1^ within 9 min at 70 °C. It can be seen more intuitively from the plots of polymer yield against polymerization time conducted at 10, 20, 30, and 50 °C, as plotted in Figure 1. The polymer yield increased linearly with the extension of polymerization time at the initial stage of polymerization, suggesting a steady-state stage at the beginning of the polymerization. Moreover, an increase in polymerization temperature resulted in a shorter duration to reach this steady state, implying that a higher temperature accelerates the formation of active sites and increases the rate constant for propagation. The superior robustness of the catalyst to high temperatures has effectively solved the critical deficiency of iron-based catalyst deactivation at high polymerization temperatures [35,49,50].

### 3.2. Quasi-Living Polymerization of IP

To clarify the quasi-living polymerization characteristics of the Fe(acac)_3_/Al*^i^*Bu_3_/IITP catalyst system, the evolution of Mn and PDI of the PI was monitored at the given temperatures of 30 and 50 °C. The polymerization kinetics were investigated under conditions identical to Run 2 in Table 1. The molecular weights of the obtained polymer exhibited a linear increase in polymer yields while maintaining a narrow PDI (PDI = 1.81–2.11) at each polymerization temperature, as depicted in Figure 2a,b. Accurately extrapolate the drawn line to the origin, and only observe the induction stage of polymerization at 30 °C. The linearity passing through the origin point at 50 °C suggests that the termination reaction could be ignored during the polymerization process. Furthermore, the GPC peaks corresponding to the produced PIs gradually shifted towards higher-molecular-weight regions with increasing polymer yield while retaining the unimodal shape in the upper regions depicted in Figure 2a,b. In addition, the catalytic system can still stably initiate IP polymerization through the seeding polymerization after being left for 8 h under the condition of IP deficiency (Appendix A), indicating the living and controllable nature of the polymerization process [51].

The semi-logarithmic plots of ln([M]_0_/[M]_t_) ([M]_0_ represents the initial concentration of the monomer, and [M]_t_ denotes the monomer concentration at the specified time) against polymerization time as depicted in Figure 2c were all linear (R^2^ = 0.953~0.988). These results support a constant active center concentration throughout the polymerization process, and irreversible chain transfer and chain termination reactions can be neglected. A true-first-order rate constant *k*_obs_ of 0.0004, 0.0015, 0.0098, and 0.3033 L mol^−1^ min^−1^ were calculated for each polymerization at 10, 20, 30, and 50 °C, respectively. This means the elevation of polymerization temperatures can accelerate the polymerization rate and shorten the steady-state duration of polymerization. The linear relationships between 1/T × 10^3^ and −ln*k* are shown in Figure 2d, revealing a relatively high activation energy (*E*) of 92.78 kJ/mol as calculated by the Arrhenius equation in the iron-catalyzed conjugated diene co-ordination polymerization system, indicating that IP polymerization induced by the Fe(acac)_3_/Al*^i^*Bu_3_/IITP system can proceed faster or even more efficiently at higher temperatures.

### 3.3. Soft–Hard Block Copolymerization of 3,4-PI and s-1,2-PBD

Our previous study demonstrated the quasi-living polymerization characteristics of the current catalyst system toward BD polymerization [42]. Therefore, selecting BD as the second monomer and synthesizing soft and hard block 3,4-PI-*b*-*s*-1,2-PBD through sequential polymerization has been proven feasible, and the experimental route and results are listed in Table 2. For the one-pot sequential polymerization process, the first stage of polymerization was conducted at [IP]/[Fe] = 500 at 30 °C until the IP was totally consumed; a molecular weight of *M*_n_ = 19.9 × 10^4^ with a PDI of 2.02 was obtained (Figure 3, right GPC curve). Subsequently, BD was rapidly added into the polymerization system to initiate the second stage of polymerization at [BD]/[Fe] ratios of 500, 750, 1000, or 1500 for an additional duration of 4 h. The GPC curve of the copolymer gradually shifts to a higher region compared to that of the former IP homopolymer while remaining with unimodal traces and a slightly broadened molecular weight distribution, suggesting a genuine PI and PBD block copolymer was successfully formed rather than a mixture of two homopolymers (Figure 3, left GPC curve).

The structures of the copolymers were confirmed by the ^1^H and ^13^C NMR spectra (Figure 4). As shown in Figure 4a, the peak located at 5.45 ppm in the ^1^H NMR spectrum was assigned to the -C***H***=CH_2_ (p) and -C***H***=C***H***- (l) protons of the 1,2 units and 1,4 units in the PB segment, respectively. The intensity of these peaks increased with an increase in the BD feed ratio during polymerization. Based on the relative peak intensities of -C=C***H***_2_ (d), -C=C***H***- (e), and -C***H***=CH_2_ (j) located at 4.84, 5.23, and 5.84 ppm, the representative sample (Run 20, Table 2) provides a polyisoprene segment content of 54.2% consistent with the feed ratio of IP, comprising of approximately 50.3% of 3,4, 46.2% of 1,4, and 3.5% of 1,2 units. The olefin carbon region of the ^13^C NMR spectra was further used to determine the sequence structure and stereoisomerism of the same polymer (Figure 4b). The signal located at 111.8 and 147.8 ppm is assigned to the 3,4 units of PI segments, while the signals at 125.3 and 135.1 ppm are assigned to the 1,4 units. The signal of the 1,2 units of PI segments should have been observed at 150.5 and 152.7 ppm, but it cannot be detected due to its low content. The block copolymer exhibits clear peaks around 114.9 ppm and 143.0 ppm, attributed to -CH=***C***H_2_ and -***C***H=CH_2_ in 1,2 units of the *s*-1,2-PBD segments, respectively. The chemical shifts of the syndiotactic pentad *rrrr* was around 114.6 ppm, and the calculated *rrrr* content of *s*-1,2-PBD was 35.5% (Run 20, Table 2). The *rrrr* content increased with the increment of the BD feed ratio (*rrrr* are shown in Table 2), in a range of 35.5% to 48%. The results are consistent with our previously reported results [25]. The H-C bond correlations of the block copolymers were further determined through ^1^H-^13^C HSQC (Figure 4c). For the PI block, the cross-peak of *δ*_C_/*δ*_H_ at 4.84/111.8 ppm is identified for the allyl group of the 3,4 units, and the cross-peak of *δ*_C_/*δ*_H_ at 5.23/125.3 ppm is identified for the C=C bonds of the 1,4 units. The cross-peaks of the 1,2 units in the PBD block are visible at the *δ*_C_/*δ*_H_ of 5.05/114.9 and 5.45/143.0 ppm for the vinyl group. Additionally, the cross-peak at 5.34/129.1 ppm belongs to the C=C bonds of the 1,4 units in the PBD block.

The thermal properties of the 3,4-PI-*b*-*s*-1,2-PBD diblock copolymers were further investigated by DSC analysis. As shown in Figure 5a, only *T*_g_ was observed on the spectral lines of 3,4-PI, further confirming that the IP product produced by the catalytic system used is amorphous. The copolymers obtained by the in situ addition of BD possessed both *T*_g_ and *T*_m_, which were close to the values for the typical 3,4-PI and the *s*-1,2-PBD homopolymers. Moreover, with increasing *s*-1,2-PBD blocks, gradually decreasing the glass transition temperature of the copolymer from −17.5 °C to −18.3 °C was clearly observed. It is worth noting that the melting point of block copolymers surpasses that of the *s*-1,2-PBD homopolymer. This phenomenon may be due to the confinement-driven melting by a negative pressure imposed on the crystals by the neighboring amorphous regions [52]. The XRD diffraction pattern in Figure 5b shows the block copolymers at 2*θ* value of 13.7° (010), 16.5° (200/110), 21.5° (210), 23.4° (110/210), and 28.3° (120) after adding the second segment of BD feedings, which can be attributed to the *s*-1,2-PBD sequences [53,54], and the intensity of the peaks increased with the syndiotactic products. The results above demonstrate the successful initiation of BD monomer polymerization by the living iron-PI chain end generated in the first step.

The morphology of the 3,4-PI-*b*-*s*-1,2-PBD diblock copolymers was characterized using TEM, along with *s*-1,2-PBD and a blend of 3,4-PI and *s*-1,2-PBD for a comparative analysis. As shown in Figure 6a–c, the block polymers present two different phase domains; e.g., the hard segments (*s*-1,2-PBD) aggregated to form rodlike domain and as a connecting point distributed in the polymer matrix homogeneously, whereas the amorphous 3,4-PI block act as the polymer matrix and wraps around the hard segment phase due to the covalent bonding with the *s*-1,2-PBD block. This microphase separation in the semicrystalline block copolymer may be due to the incompatibility between the crystalline regions of the *s*-1,2-PBD blocks and amorphous 3,4-IP blocks [55,56]. It is known that the morphology of the block copolymer depends on the length of each block [57]. Increasing the *s*-1,2-PBD segment content results in incrementing the quantity and size of the rodlike domain corresponding to the dispersed phase. However, the arrangement is not a perfect light and dark alternating stripe structure, which may be because of the contortion of spherulites formed by *s*-1,2-PBD segments and constraints on the movement of flexible 3,4 segments during the self-assembly process. Blending experiments with pure components provide evidence that, in the case of blends of the homopolymer 3,4-IP and *s*-1,2-PBD, the crystalline regions of *s*-1,2-PBD were observed to be embedded in the continuous phase of 3,4-IP with hundreds of nanometer-size scales. The distribution of the disperse phase appeared to be inhomogeneous, suggesting a partial compatibility between 3,4-IP and *s*-1,2-PBD (Figure 6e).

To investigate the property advantages of 3,4-PI-*b*-*s*-1,2-PBD block copolymers, the mechanical and dynamic mechanical properties were evaluated. The stress–strain curves of the 3,4-PI, *s*-1,2-PBD, and 3,4-PI-*b*-*s*-1,2-PBD samples were plotted in Figure 7a, and the specific values of the tensile strength and elongation at break are presented in Appendix A. Except 3,4-PI, the *s*-1,2-PBD and 3,4-PI-*b*-*s*-1,2-PBD samples with different lengths of hard segments in Figure 7a exhibit the typical stress–strain behavior of thermoplastic elastomers; that is, at low strains (<40%), the stress of the *s*-1,2-PBD and 3,4-PI-*b*-*s*-1,2-PBD samples increases linearly with the strain, which is the elastic deformation area that conforms to Hooke’s law. Then, evident yielding can be seen because of the succussing intermolecular relative sliding occurrence. In the end, the specimen breaks at a relatively high strain. It can be attributed to physical crosslinking induced by the crystalline structures of *s*-1,2-PBD. In particular, the tensile strength of 3,4-PI-*b*-*s*-1,2-PBD increased from 4.75 MPa to 8.66 MPa with increasing BD content, while the elongation at break decreased from 579% to 400%. They combine the advantages of 3,4-PI and *s*-1,2-PBD, and their mechanical properties can be easily controlled by adjusting the [IP]/[BD] molar ratio in polymerization.

The tire industry aims to achieve the goal of high-performance green tires, which is to design rubber materials with a high wet-skid resistance, low rolling resistance, and reduced abrasion loss. However, the optimization of these goals often conflicts with each other, forming a predicament known as the “devil’s triangle” [58,59]. Therefore, the dynamic mechanical properties of the 3,4-PI, *s*-1,2-PBD homopolymer, and the representative 3,4-PI-*b*-*s*-1,2-PBD (Run 20, Table 2) samples were further examined by DMA. As shown in Figure 7b, the tan δ peaks of s-1,2-PBD, the copolymer samples, and 3,4-PI monotonically increased and *T*_g_ gradually shifted toward higher temperatures over a wide temperature range around 0 °C, which is consistent with the DSC test results. At approximately 60 °C, 3,4-PI-*b*-*s*-1,2-PBD exhibited tan δ values intermediate between those of the 3,4-PI and *s*-1,2-PBD homopolymers, and its value was closer to that of 3,4-PI. The evaluation of the wet-skid resistance and rolling resistance performance of tires during exercise depends on the tan δ of DMA at 0 °C and 60 °C, respectively, where a higher tan δ at 0 °C signifies an enhanced wet-skid resistance, and a lower tan δ at 60 °C reflects a decreased rolling resistance [60]. The tan δ values were collected in Appendix A. The 3,4-PI-*b*-*s*-1,2-PBD copolymer inherited the higher values (up to 1.157) and lower values (as lower 0.047) of 3,4-PI at 0 °C and 60 °C, implying that the introduction of 3,4-PI can effectively enhance the fuel efficiency and wet-skid resistance of *s*-1,2-PBD, and the high modulus of *s*-1,2-PBD can compensate for the poor wear resistance of 3,4-PI. Consequently, the synthesized 3,4-PI-*b*-*s*-1,2-PBD copolymer system comprising soft and hard segments emerges as a promising candidate for enhancing tire performance.

## 4. Conclusions

In this study, the isoprene polymerization behavior using robust Fe(acac)_3_/Al*^i^*Bu_3_/IITP was comprehensively characterized. The catalyst system exhibited extremely high activity (5.43 × 10^6^ g mol_(Fe)_^−1^ h^−1^) and excellent thermal stability for IP polymerization. The resulting polymers had a high molecular weight and major 3,4 (+1,2) selectivity. Kinetic studies demonstrated that the system could catalyze IP polymerization in a quasi-living manner, and high temperature favors efficient polymerization. Moreover, based on this quasi-living polymerization characteristic, an in situ initiated soft–hard block copolymer, 3,4-PI-*b*-*s*-1,2-PBD, a product with a controllable block content consisting of amorphous 3,4-PI segments and crystalline *s*-1,2-PBD segments was successfully synthesized. The microphase separation in semicrystalline block copolymers is attributed to the incompatibility between the crystallization of the *s*-1,2-PBD block and the amorphous regions within the 3,4-IP block. Furthermore, the introduction of *s*-1,2-PBD is expected to endow copolymers with a high modulus without sacrificing the high wet-skid resistance and low rolling resistance of 3,4-PI. This work proposes, for the first time, an approach for preparing 3,4-PI-*b*-*s*-1,2-PBD soft–hard block copolymers using a novel catalytic system Fe(acac)_3_/Al*^i^*Bu_3_/IITP. This work enriches the theory and practice of co-ordination polymerization from a methodological perspective and provides an effective way to fabricate high-performance synthetic rubber materials.

## Figures and Tables

**Figure 1 polymers-16-01172-f001:**
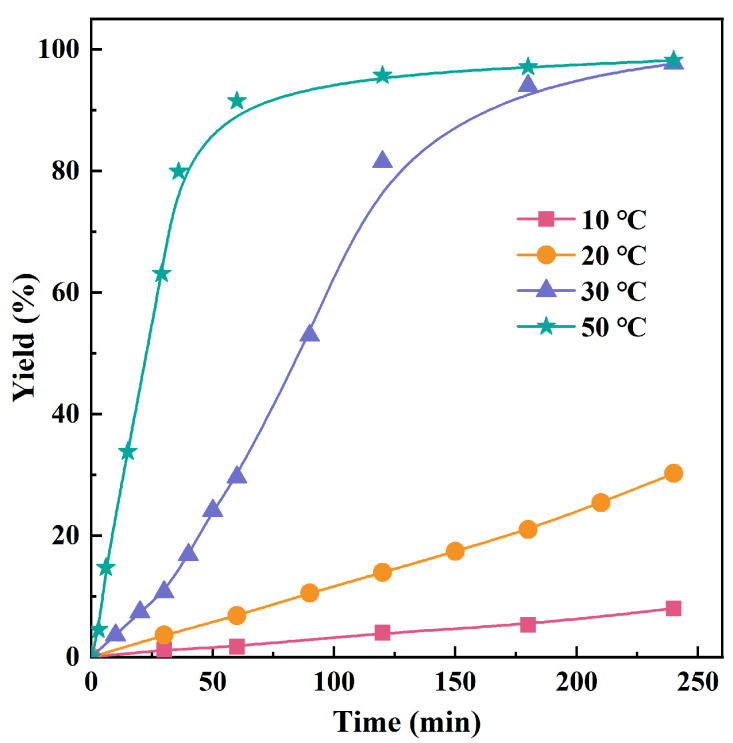
The evolution of polymer yield on the time at various temperatures.

**Figure 2 polymers-16-01172-f002:**
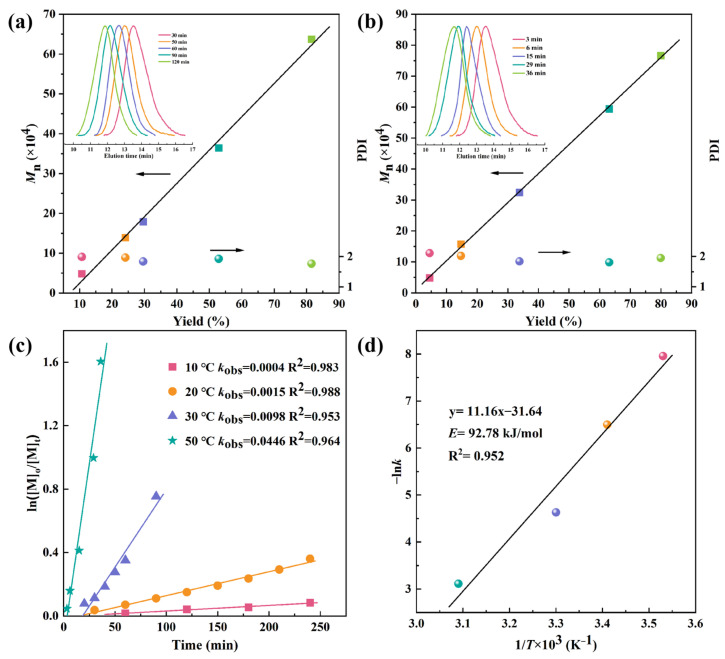
Plots of *M*_n_ and PDI as a function of the yield of IP at (**a**) 30 °C and (**b**) 50 °C; (**c**) first-order plots of polymerizations of IP at different temperatures; (**d**) Arrhenius plot for the polymerization of IP.

**Figure 3 polymers-16-01172-f003:**
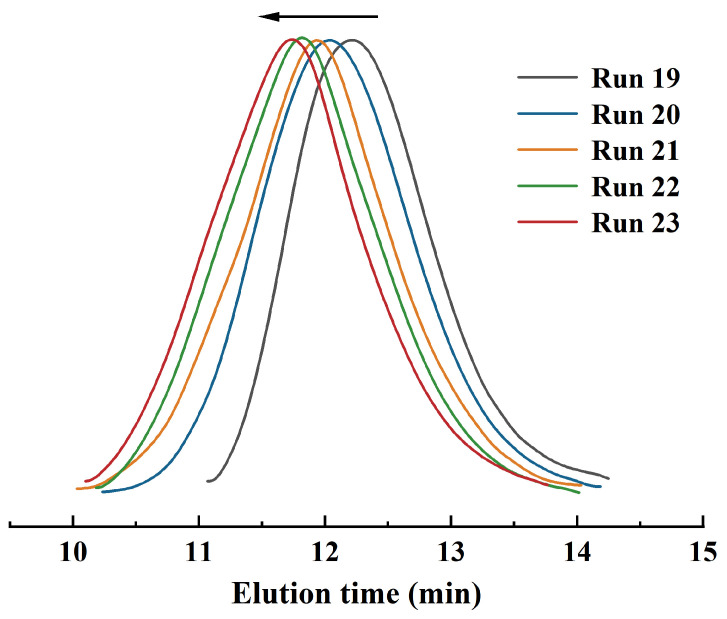
GPC profiles of final 3,4-PI-*b*-*s*-1,2-PBD copolymers (Runs 20–23) and corresponding 3,4-PI before feeding BD (Run 19).

**Figure 4 polymers-16-01172-f004:**
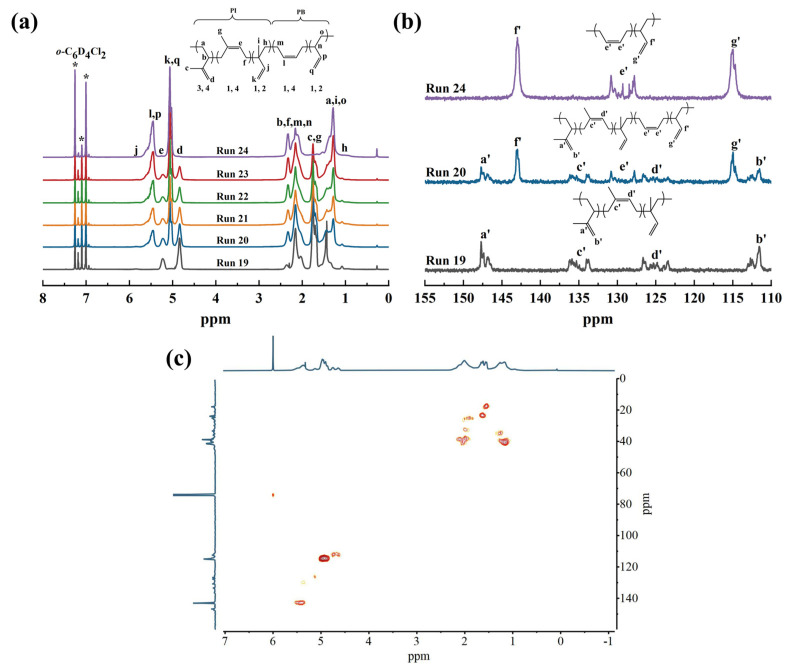
(**a**) ^1^H NMR, (**b**) ^13^C NMR, and (**c**) ^1^H-^13^C HSQC spectra of the resultant 3,4-PI (Run 19), 3,4-PI-*b*-*s*-1,2-PBD copolymers (Run 20–23), and *s*-1,2-PBD (Run 24).

**Figure 5 polymers-16-01172-f005:**
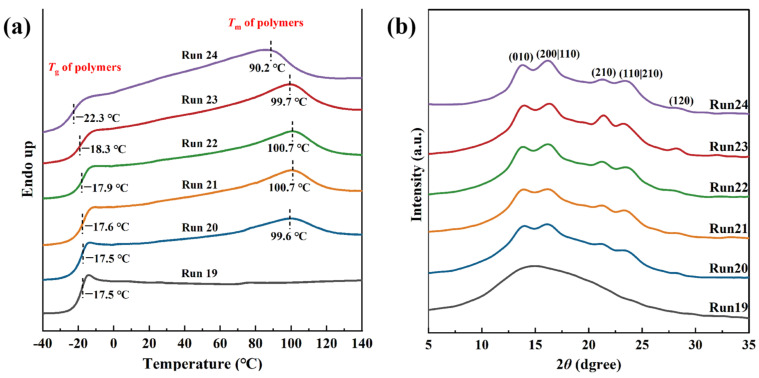
(**a**) DSC and (**b**) XRD profiles of the 3,4-PI (Run 19), 3,4-PI-*b*-*s*-1,2-PBD copolymers (Run 20–23), and *s*-1,2-PBD (Run 24).

**Figure 6 polymers-16-01172-f006:**
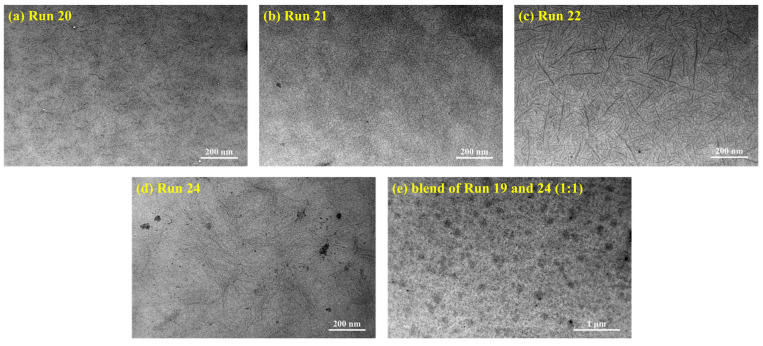
TEM images of 3,4-PI-*b*-*s*-1,2-PBD copolymers: (**a**) Run 20, (**b**) Run 21, (**c**) Run 22; (**d**) Run 24; and (**e**) blend of Runs 19 and 24 (1:1).

**Figure 7 polymers-16-01172-f007:**
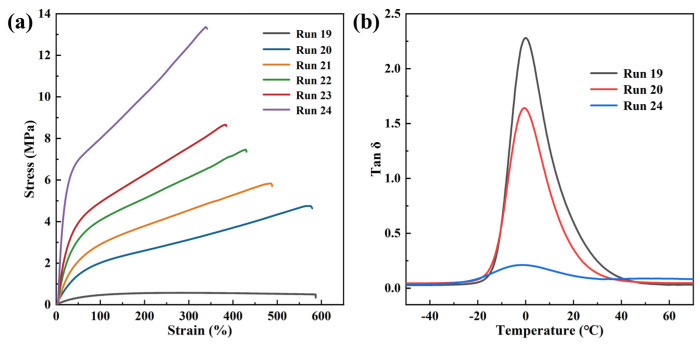
(**a**) Stress-strain curves of 3,4-PI (Run 19), 3,4-PI-*b*-*s*-1,2-PBD copolymers (Run 20-23), and *s*-1,2-PBD (Run 24); (**b**) Representative tan δ curves versus temperature of 3,4-PI (Run 19), 3,4-PI-*b*-*s*-1,2-PBD (Run 20), and *s*-1,2-PBD (Run 24) at 0.1% tensile strain.

**Table 1 polymers-16-01172-t001:** Effect of catalyst component under various conditions on the behavior of IP polymerization *^a^*.

Run	Cocat.	[Al]/[Fe]	[IP]/[Fe]	Yield (wt%)	*M*_n_^*b*^ (×10^4^)	PDI *^b^*	Microstructure *^c^* (%)			*T*_g_^*d*^ (°C)
							3,4	1,2	1,4	
1	Al*^i^*Bu_3_	10	15,000	94.6	73.1	1.97	51.6	3.0	45.4	−16.6
2	Al*^i^*Bu_3_	20	15,000	97.7	91.3	1.92	51.2	3.2	45.6	−15.8
3	Al*^i^*Bu_3_	30	15,000	97.1	82.5	1.92	51.1	3.1	45.8	−17.6
4	AlEt_3_	10	15,000	trace	--	--	--	--	--	--
5	AlEt_3_	20	15,000	95.8	65.0	2.03	51.1	2.6	46.3	−17.2
6	AlEt_3_	30	15,000	88.3	75.5	1.90	52.1	3.0	44.9	−16.6
7	Al*^i^*Bu_2_H	10	15,000	15.1	45.0	2.01	51.8	5.2	43.0	−14.1
8	Al*^i^*Bu_2_H	20	15,000	95.9	64.5	2.03	48.8	5.0	46.2	−16.0
9	Al*^i^*Bu_2_H	30	15,000	90.4	69.0	2.20	49.4	5.1	45.5	−16.7
10	Al*^i^*Bu_3_	20	10,000	99.8	70.2	1.94	51.2	2.6	46.2	−17.3
11	Al*^i^*Bu_3_	20	30,000	72.0	100.3	1.88	50.3	3.7	46.0	−16.3
12	Al*^i^*Bu_3_	20	50,000	42.0	102.7	1.88	50.8	3.9	45.3	−16.1

*^a^* Polymerization conditions: in hexane at 50 °C for 4 h, [IP] = 2.3 mol L^−1^, [P]/[Fe] = 3 (mol/mol); *^b^* Determined by GPC (using polystyrene as calibration); *^c^* Determined by NMR; *^d^* Determined by DSC; “--” not determined).

**Table 2 polymers-16-01172-t002:** Preparation of 3,4-PI-*b*-*s*-1,2-PBD by Fe(acac)_3_/Al*^i^*Bu_3_/IITP system ^*a*^.

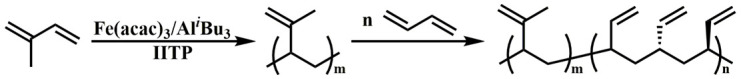
Run	[IP]/[BD] Feed Ratio	Yield(%)	*M*_n_ *^b^*(×10^4^)	PDI * ^b^*	PI Microstructure (%) ^*c*^	PBD Microstructure(%) ^*c*^	BD inCopolymer *^c^* (mol%)	*rrrr* ^*d*^ (%)	*T*_g_ ^*e*^(°C)	*T_m_ ^e^*(°C)	*X_c_ ^f^*(%)
3,4	1,2	1,4	1,2	1,4
19	500:0	100	19.9	2.02	52.1	2.1	45.8	--	--	0.0	--	−17.5	--	--
20	500:500	98.4	27.0	2.35	50.3	3.5	46.2	77.7	22.3	45.8	35.5	−17.5	99.7	10.5
21	500:750	97.4	35.8	2.43	49.3	5.0	45.8	76.9	23.1	51.9	38.1	−17.6	100.7	11.2
22	500:1000	97.2	39.7	2.51	46.9	6.1	47.0	76.5	23.5	60.7	44.4	−17.9	100.7	16.9
23	500:1500	99.0	46.1	2.44	46.8	5.6	47.6	77.9	22.1	68.6	48.0	−18.3	99.7	18.1
24	0:500	96.3	12.4	2.46	--	--	--	64.9	35.1	100	68.0	−22.3	90.2	24.1

*^a^* Polymerization conditions: in hexane at 30 °C, [P]/[Fe] = 3, and [Al]/[Fe] = 20 (mol/mol); he first polymerization stage: [IP] = 0.5 mol L^−1^ for 3 h; the second polymerization stage: [BD] = 0.5 mol L^−1^ for 4 h; *^b^* Determined by GPC (using polystyrene as calibration); * ^c^* Determined by NMR; *^d^ rrrr*: syndiotactic index, percentage of syndiotactic pentads; ^*e*^ Determined by DSC; *^f^* relative crystallinity, calculated by the crystallinity enthalpy referred to the full value (60.7 J g^−1^).

## Data Availability

The data presented in this study are available upon request from the corresponding author (due to privacy).

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
