# Peer review of "Preparation and Characterization of Soft-Hard Block Copolymer of 3,4-IP-b-s-1,2-PBD Using a Robust Iron-Based Catalyst System"

_polymers, 2024, doi:10.3390/polym16081172_

Round 1

Reviewer 1 Report

Comments and Suggestions for Authors

This manuscript explores the use of a new catalyst system for coordination polymerization of isoprene and isoprene-butadiene block copolymers. A comprehensive set of characterization techniques were performed to understand the reaction and demonstrate living polymerization. Overall, this work is well done and this reviewer recommends publication after minor revisions.

Line 86: Empty space

Methods section: i) The solvent appears to be hexane, which is highly volatile. This combined with heat results in a pressurized cell. Can the authors provide the setup or unique apparatus (if any) for high pressure reaction?

ii)  The authors should provide the typical scale of the reaction. Only the volume for the termination reagent was provided.

Line 154: It would be helpful if the authors showed a schematic of the interactions between cocatalyst Fe(acac)3/ AliBu3/IITP system and where does the monomers coordinate.

Line 205: The authors showed living polymerization and argued that chain transfer can be ignored. Does this imply that the catalyst remains coordinated to the polymer at the end of polymerization, hence, exist as an end group? If so, how would this affect the mechanical properties and stability of the polymer? This should be clarified in the text.

Line 259: The proton NMR spectra is critical for determining tacticity here but it appears to have many overlaps because the groups are chemically similar. This reviewer recommends 2D NMR HSQC to get proton-carbon correlation.

Figure 6: Were the block copolymers stained to obtain the contrast observed in TEM? If so, what was used, and which phase do we expect the dark phase to be? Considering both blocks have similar chemistry, it would be challenging to distinguish them.

Figure 7b: In the captions, The block copolymer is labelled Run 22 but the Figure legend displays Run 20.

Author Response

Thank you for your valuable comments and suggestions. We have carefully considered your comments and revised the manuscript accordingly. The following content affords our point-to-point responses to your comments and the revised manuscript. We appreciate your kind consideration to our revised manuscript.

Reviewer 2 Report

Comments and Suggestions for Authors

The presented work has scientific novelty, and the resulting block copolymers may be of certain applied interest. The results are presented logically and consistently, the conclusions are confirmed by experimental data. At the same time, although a number of interesting results were obtained, confidence intervals were not marked on the kinetic curves. Please note the confidence intervals. It is also necessary to provide a stereoregulation mechanism, this will give the article more clarity. In Fig. 2 (blue kinetic curve) does not start from the origin, which requires explanation. Construct a mathematical relationship between the polydispersity parameter and time. As a rule, the polydispersity parameter for living polymerization processes is significantly less than two, how do the authors explain the significant PDI values?

Overall the work is good and may be considered for publication after major revision. 

Author Response

(The authors gave the same response as above.)

Round 2

Reviewer 2 Report

Comments and Suggestions for Authors

The authors have adequately corrected the manuscript and now it can be accepted.